# Exposure of French Children and Adolescents to Advertising for Foods High in Fat, Sugar or Salt

**DOI:** 10.3390/nu13113741

**Published:** 2021-10-23

**Authors:** Hélène Escalon, Didier Courbet, Chantal Julia, Bernard Srour, Serge Hercberg, Anne-Juliette Serry

**Affiliations:** 1Santé Publique France, French National Public Health Agency, 94415 Saint-Maurice, France; Anne-Juliette.SERRY@santepubliquefrance.fr; 2IMSIC, Aix-Marseille University, 13005 Marseille, France; didier.COURBET@univ-amu.fr; 3IMSIC, University of Toulon, 83041 Toulon, France; 4Sorbonne Paris Nord University, Inserm U1153, Inrae U1125, Cnam, Nutritional Epidemiology Research Team (EREN), Epidemiology and Statistics Research Center, University of Paris (CRESS), 93000 Bobigny, France; c.julia@eren.smbh.univ-paris13.fr (C.J.); b.srour@eren.smbh.univ-paris13.fr (B.S.); s.hercberg@eren.smbh.univ-paris13.fr (S.H.); 5Public Health Department, Avicenne Hospital, Assistance Publique des Hôpitaux de Paris (AP-HP), 93000 Bobigny, France; 6French Network for Nutrition and Cancer Research (NACRe Network), 78352 Jouy-en-Josas, France

**Keywords:** food marketing, food advertising, children, Nutri-Score, television, low nutritional quality, high in fat, salt and/or sugar

## Abstract

Food marketing of products high in fat, sugar and salt (HFSS), including television advertising, is one of the environmental factors considered as a contributor to the obesity epidemic. The main objective of this study was to quantify the exposure of French children and adolescents to television advertisements for HFSS products. TV food advertisements broadcast in 2018 were categorized according to the Nutri-Score of the advertised products. These advertisements, identified according to the days and times of broadcast, were cross-referenced with audience data for 4- to 12-year-olds and 13- to 17-year-olds. More than 50% of food advertisements seen on television by children and adolescents concerned HFSS products, identified as classified as Nutri-Score D and E. In addition, half of advertisements for D and E Nutri-Score products were seen by children and adolescents in the evening during peak viewing hours, when more than 20% of both age groups watched television. On the other hand, during the same viewing hours, the percentage of children and adolescents who watched youth programs, the only programs subject to an advertising ban, was very low (<2%). These results show that the relevance of regulating advertising at times when the television audience of children and adolescents is the highest and not targeted at youth programs, in order to reduce their exposure to advertising for products of low nutritional quality.

## 1. Introduction

The prevalence of overweight and obesity in children and young people has increased significantly worldwide. The number of obese children and adolescents aged 5 to 19 has increased tenfold over the past four decades [1]. This evolution varied nevertheless from country to country. In Europe in the last two decades, an increase in the prevalence of overweight and obesity in children was observed in Mediterranean countries, while these rates have stabilized in other countries, including France [2]. More precisely, in France, the prevalence of overweight and obesity in children increased at the end of the 1990s [3], and has stabilized since the 2000s [3,4,5,6]. This prevalence nevertheless remains high, and social inequalities are marked [5,7,8].

The harmful effects of childhood obesity have been indeed widely demonstrated [9]. Obese children are at a higher risk of obesity in adulthood [10], and childhood obesity increases the risk of mortality and premature death [11,12], as well as the risk of developing chronic diseases in adulthood [13].

Obesity is a multifactorial chronic disease. Its multiple determinants—systemic, environmental (such as food marketing) and behavioral (such as low levels of physical activity and high sedentary behavior)—have been summarized in a global framework [14]. Contributing factors of obesity are also an increase in calories ingested, especially via ultra-processed foods [15,16], and more generally, an excessive consumption of sugar-sweetened beverages and of energy-dense foods with high levels of saturated fatty acids, added sugars, and/or salt [17,18] (hereinafter referred to as high in fat, salt and sugar (HFSS) foods).

Marketing strategies employed by the food industry encourage the purchase and consumption of these HFSS foods, wherein a combination of sugar, salt, fat as well as texture is proven to be highly appealing [19]. These foods have been recognized as an important factor in the excessive consumption of these products, especially by children [18]. The harmful effects on children’s health of food marketing, which includes commercial communications (defined as the transmission of any form of content (messages, images, graphic elements, symbols) intended to promote, directly or indirectly, goods, services, or the image of organizations)) [20], including television advertising, and TV sponsorships, have been highlighted in several literature reviews [21,22,23,24,25,26,27] and international organization reports [18]. Food marketing aimed at children is associated in particular with higher calorie intake after watching advertisements (television or online games), which is not compensated for by a lower calorie intake later in the day [28]. It is also associated with a higher consumption of HFSS foods, a lower consumption of healthy foods [29,30], and a higher body weight [31,32]. The impact of food marketing on food intake in children specifically has been demonstrated in a systematic review and meta-analysis. This one has shown that exposure to advertising for unhealthy food increased food intake in children under 18 years old, but not in adults [27].

Numerous studies and literature reviews on the effects and impact of food marketing have thus been carried out for several decades providing a solid scientific basis for the advocacy of international bodies seeking to limit food marketing to children [18,33,34,35,36,37]. Conducting studies to monitor the marketing of unhealthy products to children and adolescents is also fundamental to encouraging and sustaining the implementation of regulations and restrictions by States, as mentioned by the WHO regional office for Europe.

In France, relatively few studies exist on the food marketing subject [38,39,40], even though several French bodies have pleaded to ban marketing for products of low nutritional quality targeting children [41,42,43,44]. The results from the monitoring study presented here help to fill this gap and expose the French situation regarding two levers of food marketing for which we have data: television advertising and television sponsorship.

The main objective of this study was to quantify the exposure of French children and adolescents to television advertising and TV sponsorship for food products, and in particular for high in fat, salt and sugar (“HFSS”) foods, in 2018. The second objective was to quantify the exposure of children and adolescents to “youth” television programs, which are the subject of a ban on advertising on public channels [45], and self-regulatory measures in the food industry. An initial overview of food advertising investments and media usage times was realized to identify the media which is the most relevant on which to conduct the study.

## 2. Methods

The method used was based on the one of a study conducted in nine European countries for the European Commission [46]. This European study was commissioned as part of the European Joint Action on the implementation of validated best practices on nutrition (JA Best-ReMaP, 2020), one of the objectives of which is to develop a harmonized European approach to reduce the exposure of children and adolescents to food marketing, and to use common tools to monitor this exposure. France is participating in the joint action.

The same age groups as in the European study were used for comparability purposes, for children and adolescents: 4–12 years and 13–17 years. The analyses covered the year 2018.

For ease of reading, the term “advertisements” will be used to refer to the advertisements and television sponsorship analyzed in this study.

### 2.1. Source of Data

Food advertising data for 2018 (identification of advertised products and advertising investment) were collected by Kantar Media. This company identified the advertisements broadcast on all media and partially on the Internet (mainly banners on Internet sites because at the time of the study; there was no census of ads broadcast on mobile and social networks).

Data on the media usage times of children and adolescents were collected via two surveys. Data relating to the use of television were measured via the Médiamat study, conducted by the Médiamétrie Institute, carried out each year with a panel of around 5000 households that are representative of households equipped with TV, i.e., more than 12,400 individuals aged 4 and over. The data are collected automatically via an audience meter, which collects the time spent in front of each television program by each panelist, in particular according to their socio-demographic profile. Médiamétrie data relating to the age groups 4–12 years and 13–17 years were collected for this study.

The data on the press, radio, and Internet come from the Junior Connect survey, an online quota survey conducted by the Ipsos Institute on 4500 children and adolescents aged 1 to 19 in 2018.

To quantify the exposure of individuals to television advertisements for HFSS foods, two types of data were matched:2018 audience data for the two age groups studied collected via Médiamétrie’s audience meter (see above)data from Kantar Media allowing the identification of advertisements for products and food brands broadcast each day of 2018 and for each time of the day when an advertisement was broadcast.

Thus, the content of a food advertising spot (name and/or brand of the advertised food product, e.g., Kinder Bueno) broadcast on a given day, at a given time, was identified by Kantar Media, and Médiamétrie measured, in parallel, the number of people watching television at the time that the spot in question was broadcast.

### 2.2. Coding of Advertisements According to the Nutri-Score of Advertised Products

Each advertising spot or television sponsorship operation broadcast on television in 2018 identified by Kantar Media was coded according to the nutritional quality of the advertised product.

To do this, the nutritional profile on which the Nutri-Score is based, a front-of-package nutrition labelling that France adopted in 2017 that displays information about the simplified nutritional quality of pre-packaged foods and non-alcoholic drinks, was used.

The Nutri-Score is based on the nutritional profile system of the Food Standards Agency, which was originally developed and approved in the UK, where it was used to regulate advertising (OFCOM) [47,48,49]. This nutritional profile has been adapted for France by the French High Council of Public Health and named the FSAm/HCSP [50,51]. Based on the nutritional quality of a food product, the Nutri-Score uses a scale of 5 colors associated with letters going from A to E, A corresponding to the best nutritional quality, and E indicating the worst. For every 100 g or 100 mL of a product, the score measures the level of nutrients and foods the system seeks to promote (fiber, protein, fruits and vegetables, legumes, nuts, olive oil, rapeseed oil, and walnut oil), as well as the level of nutrients that it seeks to limit (calories, saturated fatty acids, sugars, and salt).

In our study, HFSS advertised products were coded with a Nutri-Score of D or E. The products were coded in three categories: “ABC Nutri-Score”, “DE Nutri-Score”, “Not classified/not applicable”. Some products a priori not concerned by the Nutri-Score were reclassified. Fresh meats, fresh vegetables, fresh fruits, coffee, teas and herbal teas, and sweeteners were not considered to be HFSS foods: an ABC Nutri-Score was assigned to them. Honey and salt, unprocessed products that do not fall under the purview of the Nutri-Score system, were classified in the DE category. The products for which it was not relevant or not recommended to apply a Nutri-Score (meal replacements, infant nutrition, yeasts, and flavourings) were coded as “not applicable”. The products and the brands or ranges that could not be associated with a Nutri-Score were coded as “Not classified”.

To code the advertised food products, the database *Open Food Facts* (http://world.openfoodfacts.org/, accessed on 18 December 2019) was mainly used. This open access collaborative database lists pre-packaged manufactured food products purchased in stores and provides, among other data, their nutritional composition and Nutri-Score. If the information on the Nutri-Score of the advertised product was not available on the *Open Food Facts* database, the brands’ websites were consulted in order to collect the nutritional composition of the products per 100 g, and to calculate their Nutri-Score. For the products advertised in the Catering family, those in the fast-food sector were either not classified, or were coded when it was possible to calculate the Nutri-Score from the nutritional composition of the promoted product (e.g., McDonald’s 280 Bacon Promo). When the full menus were advertised, they were classified according to the main component of the menu (sandwich, salad, etc.). The products advertised as “non-fast-food” restaurants (e.g., Bistrot Régent and Planet Sushi) were coded as “Not-classified”, because their products were too diverse.

In total, 1172 food products that were the subject of at least one advertisement broadcast in 2018 across all media were coded according to their Nutri-Score (ABC Nutri-Score vs. DE Nutri-Score). Regarding television specifically, 903 products that were the subject of at least one television advertisement in 2018 were coded.

### 2.3. Variables and Statistical Analyses

Four types of analyses were realized to respond to the two objectives of the study; namely, the quantification of the exposure of children and adolescents to advertising for HFSS foods and to youth TV programs:

1. Data on food investments were collected to understand the scope and relevance of our study in relation to

the media considered: did the food investments dedicated to television constitute a substantial part of the food investments made in all media?the types of products studied: how did the proportion of food advertising investments for products of low nutritional quality coded DE Nutri-Score compare to all food investments?

Two variables were calculated to study these questions:The share of television food investments out of all food investmentsThe distribution of food investments according to the Nutri-Score of the advertised products.

2. The quantification of media usage time was performed to identify the most watched by children and adolescents. This was carried out through a descriptive analysis of daily time spent in front of different media.

A descriptive analysis of children and adolescents’ television audience by time slot allowed us to identify the most watched time slots. These results were used to analyze the exposure of children and adolescents to food advertising according to this criterion (see below).

3. Quantification of the exposure of children and adolescents to television youth programs was carried out through (a) a descriptive analysis of the TV times of children and adolescents in 2018 according to the types of programs and (b) by an analysis of their audience by program time slot.

4. To quantify the exposure of children and adolescents to advertisements for HFSS foods, defined by the number of advertisements for HFFS foods seen by children and adolescents over a given period, descriptive analyses were carried out using the Mxplorer software, media performance analysis software purchased from Espaces TV.

TV food advertisements coded according to the Nutri-Score and identified according to the days and times of broadcast, were cross-referenced with audience data for 4- to 12-year-olds and 13- to 17-year-olds. An analysis of this cross-referencing by the time slot of child and adolescent audiences was been performed.

The following variables were calculated:the percentage of television advertisements for DE Nutri-Score products among food advertisements viewed in each age group studied;the audience per time slot for each age group studied, i.e., concerning children, the average percentage of children, in the French child population, who watched television during a given time slot;the percentage of television advertisements for DE Nutri-Score products seen during the time slots where the television audience was highest among children and adolescents.

## 3. Results

### 3.1. Food Investments

Food advertising investments (including food, drink, and catering) across all media (TV, radio, cinema, outdoor advertising, press, Internet) amounted to €1.1 billion in 2018, and represented 9.3% of all estimated net investments in the advertising market for all sectors and products in France. These investments were mostly directed towards television ads (60%) and Internet ads (20%) in 2018.

In 2018, 57% of food advertising investments in all media, which could be classified according to the Nutri-Score, concerned DE Nutri-Score products, while 43% concerned ABC products.

Advertising investments for products not classified according to the Nutri-Score or not concerned represented 16% of all net food advertising investments.

### 3.2. Media Usage Time

In 2018, children and adolescents spent the most time per day on TV and on the Internet. Moreover, 4- to 12-year-olds watched television for an average of 1 h and 28 min per day, and adolescents watched it for 1 h and 12 min per day. Regarding the Internet, adolescents spent nearly twice as much time on it (2 h/day) than children (53 min) (Figure 1).

### 3.3. Focus on TV

#### 3.3.1. TV Audience by Age and Time of Day

The time slot between 7:00 p.m. and 10:00 p.m. (prime time) was the most watched by children and adolescents. During this time slot, the audience per time slot, was greater than 20% for the two age groups (Figure 2 and Figure 3).

When considering audiences greater than 10%, for children aged 4–12, two other times of the day were concerned: in the morning (between 7 a.m. and 11 a.m.) and in the middle of the day (between noon and 2 p.m.), and the evening slot widened from 4 p.m. to 11 p.m. (Figure 2). For 13- to 17-year-olds, the middle of the day was added, and the evening window was widened. More than 10% of them watched TV between 12 p.m. and 2 p.m. and in the evening between 6 p.m. and 11 p.m. (Figure 3).

#### 3.3.2. Most-Watched TV Genres and Audience for Youth Programs by Time Slots

Youth television programs were seldom watched by children and adolescents. These programs represented 0.4% of TV time seen in the year by 4- to 12-year-olds, respectively 0.1% for adolescents. More precisely, the audience of these youth programs by time slots (i.e., the percentage of individuals watching youth programs by time slots during the day) was very low among children and adolescents (less than 2%), in particular during the hours when the most minors, in terms of percentage, were watching TV (audience greater than 10%) (Table 1).

In addition. among the genres of programs. the most watched. fiction. mainly broadcast between 8 p.m. and midnight. was in first place among the three age groups studied. It represented 46.4% of total television time in 2018 for 4- to 12-year-olds. 35.0% for 13- to 17-year-olds. and 31.0% for people aged 18 years and over.

### 3.4. Exposure of Children and Adolescents to Television Advertisements on HFSS Products

The exposure of children and adolescents to television advertisements on HFSS products was measured via the percentage of television advertisements for DE Nutri-Score products viewed by age group. and according to the audience of each age group.

The food advertisements seen on television by children and adolescents were mainly advertisements for D and E Nutri-Score products. They accounted for 53.3% of food ads seen by children and 52.5% of ads seen by teenagers.

In view of our results regarding the TV audience according to time slot. three audience thresholds were considered: 10%. 15%. and 20%. showing that the more viewing time slots considered. the higher the percentage of ads for Nutri-Score products seen by children and adolescents. Overall. 87.5% of advertisements for D and E Nutri-Score products were seen at times when more than 10% of children were watching television (Figure 4). respectively 74.5% for adolescents (Figure 5);Moreover. 60.8% of ads for D and E Nutri-Score products were seen when more than 15% of children were watching television (Figure 4). respectively 59.6% for adolescents (Figure 5);Finally. 47.8% of ads for D and E Nutri-Score products were seen by children (Figure 4) (respectively 50.2% for adolescents. Figure 5) between 7 p.m. and 10 p.m.. when more than 20% of this group was watching television.

## 4. Discussion

Children and adolescents have been identified as being particularly vulnerable to food marketing [27,33]. Food advertisements regulation is a major public health measure to limit the consumption of unhealthy food products [35]. For the implementation of such a measure. precise indicators derived from scientific studies must be provided to health policy makers. The results of this study contributed to identifying such indicators.

First. as media usage time is likely to influence the number of food marketing actions (derived by food advertising investment) to which children and adolescents are exposed; a baseline for these two dimensions was first established to identify the media on which to focus.

In France. in 2018. the media with which children aged 4–12 spent the most time was television. followed by the Internet. Concerning adolescents. daily television time remained high. but lower than that spent on the Internet. Since 2012. an increase in the use of the Internet to the detriment of television for both age groups and for children in particular was observed in France [52] suggesting a continued increase in usage times for this media for both age groups in the years to come. The same trends are observed in other countries. both with regard to the predominance of television and the change in media usage times according to age [53]. Our results regarding food investment show that they were mostly allocated to television (60%) and the Internet (20%) in 2018.

These two results about media usage times and food investments would have plead to analyze the exposure of children and adolescents to advertisements for HFSS foods broadcast on TV and the Internet. The restriction to TV medium and the disregard of advertisements disseminated on websites and social networks was due to an external constraint to the study: the lack of declared data on investments and targeting made on this media to date.

The results of our study on the exposure of children and adolescents to unhealthy food advertisements focused on television for this reason.

Our data demonstrates. for the first time in France. that food advertising seen on television by children and adolescents are mainly (more than 52%) advertisements for HFSS products. classified with a Nutri-Score of D and E. In addition. half of ads for D and E Nutri-Score products are seen by children and adolescents in the evening during peak viewing hours. when more than 20% of 4- to 17-year-olds are watching television. The study also highlights notable results on television youth programs. Since 2016. these programs have been banned from advertising on French public channels (law on the removal of commercial advertising from youth programs on public television [45]). and subjected to self-regulation measures on the part of the food industry. The study shows that youth programs represent only 0.4% of TV time seen in the year by 4- to 12-year-olds (0.1% respectively for 13- to 17-year-olds). on public and private channels combined. suggesting an even lower figure for solely public channels.

In addition. at times when the percentage of children and adolescents who watch television is the highest. the percentage of children and adolescents who watch youth programs is very low (between 0.7% and 1.9%). These results. quantified for the first time in France. show the insufficiency of the only regulatory measure taken to date in the country to reduce the exposure of children to advertising for HFSS foods. They confirm the insufficiency of the single ban on advertising for HFSS foods during youth programs that had been pointed out for France but not quantified [54]. The low effectiveness of this measure had also been highlighted for other countries [18,55].

Other criteria must therefore be considered in order to regulate this issue. TV audience share. which corresponds to the distribution of viewers’ audiences by age. has been used in several countries [18]. but this indicator has strong limitations [55]. the percentage of children and adolescents among all viewers being generally low.

According to the results of our study. taking into account the audience according to the time slots of children and adolescents appears more relevant to regulate food advertising. Indeed. the largest audience of children and adolescents is observed from 7 p.m. to 10 p.m.. while very few of them watch youth programs at these times. Youth programs are indeed broadcast mainly between 6 a.m. and 9 a.m. The programs most watched by children and adolescents are in fact fiction. programs mainly broadcast between 8 p.m. and midnight; one of the time slots where the largest volume of advertising is broadcast [53]. Children and adolescents are thus much more likely to watch programs that are not intended for them than youth programs.

These results are in line with those of other countries. in particular the United Kingdom. which showed in 2019 that 55.4% of the advertisements seen between 6 p.m. and 9 p.m. by young people aged 4 to 15 were advertisements for HFSS foods [55,56]. These results have led the country to propose to regulate advertising according to time slots [57] between 5:30 a.m. and 9 p.m. [55] as was announced by the government in May 2021. New legislation is expected in April 2022.

In the same vein. and in view of the results of the present study. Santé publique France has issued recommendations for regulating television advertising for low nutritional quality products according to different time slots throughout the day [58]. One of the recommendations of the public health agency is to ban advertising. product placement. and TV sponsorship for products with a Nutri-Score of D and E and associated brands between 7 p.m. and 10 p.m.. time slots during which more than 20% of children aged 4- to 12-years-old and adolescents between the ages of 13 and 17 are watching television. Other prohibition scenarios targeting broader time slots have also been formulated [58]. Santé publique France has also included in its recommendations the regulation of advertising disseminated on the Internet. insofar as the time devoted to this medium increases rapidly for children and adolescents [58].

All of these recommendations specify and are in line with the recommendations of the WHO and those of several French public authorities [41,42,43].

Regulatory measures have been shown to be the most effective at reducing the exposure of children to food advertising compared to self-regulation initiatives carried out by actors in the food industry [35]. However. they are difficult to implement. given the economic challenges for the audio-visual and agro-food sectors and the lobbying actions implemented by these pressure groups. Thus. in France. despite the recommendations of many health authorities and the mobilization of a part of civil society. co-regulatory measures are favored [18,54]. This was the case recently. in December 2020. when the Charter of the Superior Audio-Visual Council was adopted [59]. as an advertising regulation as part of the application of the European directive on audio-visual media services of November 2018 [60].

A first limitation to our study is due to the only media platform studied. which was television. Regarding the Internet. numerous advertisements and commercial communication messages are broadcast on social networks. online video platforms. mobile applications. and digital video game platforms. To date. it is impossible to accurately estimate the level of exposure of children and adolescents to digital marketing. due to a lack of data on investments and targeting. It is therefore necessary to continue work on methodologies to better understand the exposure of minors to advertisements for HFSS foods on the Internet. given the rapid development of the use of this medium by children and adolescents. A second limitation to our study is methodological and specific to any television audience study. Just because an individual is counted in the audience does not mean that they are actually viewing the content. For example. with the rise of multi-screen viewing. users sometimes watch online via their smartphones at the same time that they watch television. However. these data are collected by an audience meter and not self-reported by individuals. For this reason. this data collection is the best that can currently be used.

## 5. Conclusions

The overall results of this study call for the restriction of food marketing. and more specifically of commercial advertising for HFFS products. especially on television. at times when the largest number of children and adolescents watch television. This would allow to avoid the viewing by minors of at least half of the food advertising for HFSS products. In addition. the predominant use of the Internet among adolescents observed in the study, and the increase in Internet use by children and adolescents identified in France, which suggests that regulation of the Internet thus seems just as necessary.

The results of this study provide scientific arguments to public decision-makers for enacting regulations to reduce the exposure of children and adolescents to advertisements for products of low nutritional quality classified with a D or E rating by the Nutri-Score system.

## Figures and Tables

**Figure 1 nutrients-13-03741-f001:**
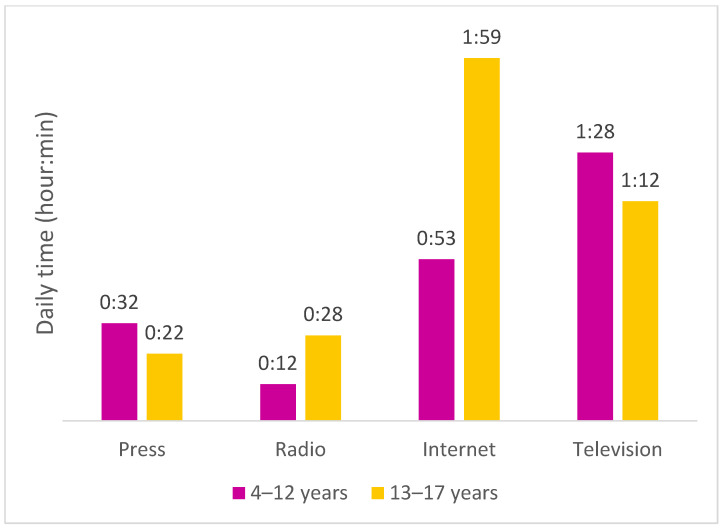
Daily time spent on different media by children and adolescents, 2018. Data source: 4- to 17-years-old: Ipsos Junior Connect 2018 (Press/Internet/Radio), Médiamétrie Médiamat (TV) 2018.

**Figure 2 nutrients-13-03741-f002:**
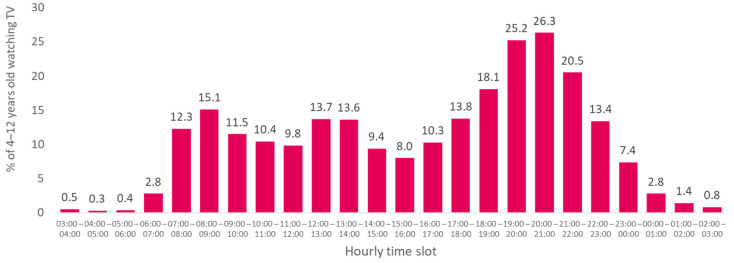
Average percentage of viewers among 4- to 12-year-olds by hourly time slot, 2018.

**Figure 3 nutrients-13-03741-f003:**
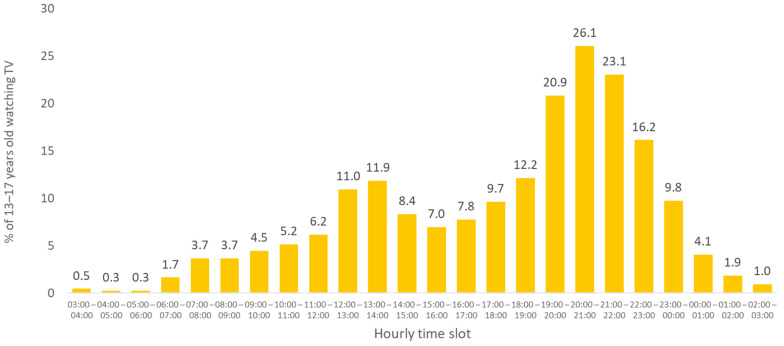
Average percentage of viewers among 13- to 17-year-olds by hourly time slot, 2018.

**Figure 4 nutrients-13-03741-f004:**
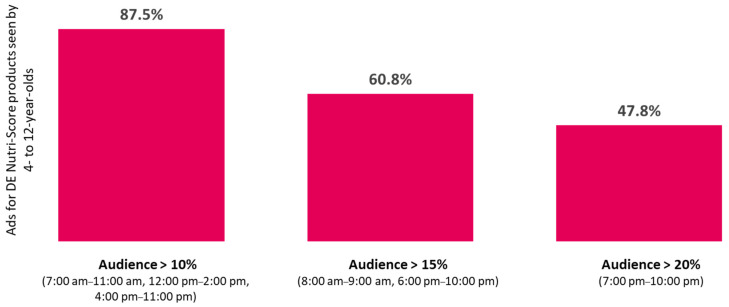
Among all the D and E Nutri-Score advertisements seen by 4- to 12-year-olds: percentages of ads shown during times when the percentage of children watching TV are above 10%. 15%. and 20%. 2018. Explanation: 87.5% of advertisements for Nutri-Score DE products were seen during time slots when more than 10% of 4- to 12-year-olds watch television. i.e., between 7 a.m. and 11 a.m. between 12 p.m. and 2 p.m. and between 4 p.m. and 11 p.m. (See Figure 2).

**Figure 5 nutrients-13-03741-f005:**
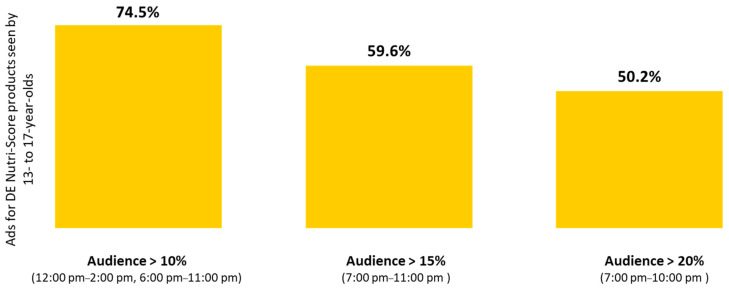
Among all D and E Nutri-Score advertisements seen by 13- to 17-year-olds: percentages of ads shown during times when the percentage of children watching TV are above 10%. 15%. and 20%. 2018.

**Table 1 nutrients-13-03741-t001:** Percentages of 4- to 12-year-olds and 13- to 17-year-olds watching youth programs compared to the percentage of those who watch television, according to time slot. Reading key: In the morning between 8 a.m. and 9 a.m., the time slot when 15.1% of children aged 4–12 watch television, 0.9% of 4- to 12-year-olds watch so-called youth programs. In bold type: Percentages observed when audiences were higher than 10%.

	Time Slots																
	07:00–8:00	08:00–09:00	09:00–10:00	10:00–11:00	11:00–12:00	12:00–13:00	13:00–14:00	14:00–15:00	15:00–16:00	16:00–17:00	17:00–18:00	18:00–19:00	19:00–20:00	20:00–21:00	21:00–22:00	22:00–23:00	23:00–00:00
% of 4- to 12-year-olds watching TV	**12.3**	**15.1**	**11.5**	**10.4**	9.8	**13.7**	**13.6**	9.4	8.0	**10.3**	**13.8**	**18.1**	**25.2**	**26.3**	**20.5**	**13.4**	7.4
% of 4- to 12-year-olds watching youth programs	**0.7**	**0.9**	**1.1**	**1.1**	0.9	**1.0**	**0.9**	0.8	0.7	**0.9**	**1.2**	**1.9**	**1.9**	**1.3**	**1.7**	**1.0**	0.4
% of 13- to 17-year-olds watching TV	3.7	3.7	4.5	5.2	6.2	**11.0**	**11.9**	8.4	7.0	7.8	9.7	**12.2**	**20.9**	**26.1**	**23.1**	**16.2**	9.8
% of 13- to 17-year-olds watching youth programs	0.1	0.1	0.2	0.3	0.2	**0.2**	**0.2**	0.2	0.2	0.2	0.3	**0.4**	**0.3**	**0.5**	**1.1**	**0.6**	0.3

## Data Availability

We used Open Food Facts, a publicly archived datasets during the study to code the food advertisements according to the Nutri-Score: https://fr.openfoodfacts.org/, accessed on 15 June 2021. Advertising investment and schedule broadcat data were purchased and include named opera-tors. They are not public datasets.

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
