# Peer review of "Exposure of French Children and Adolescents to Advertising for Foods High in Fat, Sugar or Salt"

_nutrients, 2021, doi:10.3390/nu13113741_

Round 1
Reviewer 1 Report
The problem discussed in the article is important due to the growing phenomenon of obesity. Time spent in front of the television, including watching food advertisements, can be seen as a major cause of obesity. However, the problem is more complex. Watching television may be also associated with less physical activity, which is not mentioned in the work. The use of available databases did not allow this aspect to be taken into account. A serious problem that hinders the interpretation of the results is the limitation indicated by the Authors which is specific to any television audience study. Individuals may be counted in the audience though they are not actually viewing the content.
For me, other elements are also a matter of concern:
Lines 90-92. The main objective of the study was to quantify the exposure of French children and adolescents to television advertising and TV sponsorship for food products and in particular for high in fat, salt and sugar (“HFSS”) foods in 2018. Please explain why people 18 years and over were included in the analysis. I do not agree that this age group may be used as a comparison group. Please consider changes to the title of the manuscript.
Lines 95-96. In my opinion the sentence ”An overview of food advertising investments and media usage times provides the background for the justification and progress of the study” needs to be rewritten.
The section 2.1. ‘Source of data’ should be rewritten due to an excessive amount of information. The main objective of the study was to quantify the exposure of French children and adolescents to television advertising and TV sponsorship for food products. Thus, information about Kantar Media research is not needed. For example: ‘This company identified the advertisements broadcast on television, in the cinema, on the radio, in the press, or by outdoor advertising and, in part, advertisements on the Internet’. I understood that the television advertisement is in the scope of the presented results of analysis. Moreover, information on ‘data relating to radio come from the Médiamétrie 126,000 survey, a telephone survey with quota sampling conducted each year among 126,000 people.. “ leads to misunderstanding. Short information on the data from these surveys used in the analysis will be satisfying.
Lines 257 – 262 The Figure 1 is not necessary in the manuscript. It presents only two numbers which are described earlier.
Line 266 In my opinion the word ‘media’ is not necessary in this sentence.
Figure 3, 4, and 5 – the same colors should be used to show the audience per time slot.
Lines 318-322 The section 3.4.1 is too short to be identified.
The section 3.4.2. ‘Percentage of television advertisements for DE Nutri-Score products viewed according to the audience of children and adolescents’ needs improvement. Figure 6 an 7 duplicate the information in the text and vice versa. The titles of figures and explanation sound a little strange. I suggest to delete the figures from the text.
In Discussion section there are a lot of results. Such repetition is not appropriate.
I suggest to rewrite the conclusion – more references to the results of the own study.
There are many editorial errors in the text, e.g. unnecessary dots, information in parentheses etc.
Some examples:
Lines 5-10. Capital letters in the names of institutions
Line 19 instead of ‘ should be .
Line 61 1 in my opinion it would be better to present short definition of communication in the text.
Author Response
The problem discussed in the article is important due to the growing phenomenon of obesity. Time spent in front of the television, including watching food advertisements, can be seen as a major cause of obesity. However, the problem is more complex. Watching television may be also associated with less physical activity, which is not mentioned in the work. The use of available databases did not allow this aspect to be taken into account.
>> We thank the reviewer for his comments on our manuscript. We agree that obesity is a multifactorial chronic disease and that low physical activity levels and high sedentary behaviour increase the risk of obesity. In a paper cited in the introduction of our manuscript (Swinburn 2011, ref. 31), the multiplicity of obesity determinants (systemic drivers, environmental drivers and behaviour patterns) is summarized in a framework that categorises obesity determinants. This framework reflects the complexity that you mention and both low physical activity levels and food marketing environments are identified as obesity determinants. à We have explicitly mentioned the multifactorial character of obesity in consequence. We have also mentioned physical activity and sedentary behaviour as well as food marketing.
Our study focused on food marketing environment only and did not include physical activity behaviour for the following reasons. Our study was a descriptive one ultimately aimed to provide to policy makers better indicators to regulate TV advertising than current ones. It used specific types of data - audience data and advertisements data - which are not of the same nature as individual behavioural data and therefore unfortunately can’t be directly linked to either data on physical activity or dietary behaviour. At best, connecting both types of databases would provide ecological correlations, with very limited evidence base.
A serious problem that hinders the interpretation of the results is the limitation indicated by the Authors which is specific to any television audience study. Individuals may be counted in the audience though they are not actually viewing the content.
>> This limitation is specific to any television audience study but audience data. They are collected by an audience meter and not self-reported by individuals. For this reason, this data collection is the best that can currently be used, despite the limitations. Following this comment, these have been added in the manuscript.
Lines 90-92. The main objective of the study was to quantify the exposure of French children and adolescents to television advertising and TV sponsorship for food products and in particular for high in fat, salt and sugar (“HFSS”) foods in 2018. Please explain why people 18 years and over were included in the analysis. I do not agree that this age group may be used as a comparison group. Please consider changes to the title of the manuscript.
>> We agree that adults were not the primary population target of our manuscript. They were initially included in the analysis for the following reasons.
The method used was based on the one of a study conducted on children and adults in nine European countries for the European Commission. This European study was commissioned as part of the European Joint Action on implementation of validated best practices on nutrition (JA Best-ReMaP, 2020), one of the objectives of which is to develop a harmonized European approach to reduce the exposure of children and adolescents to food marketing and to use common tools to monitor this exposure. France is participating in the joint action.
We added this precision in the manuscript.
The same age groups as in the European study, including adults, were used in the first version of the manuscript for comparability purposes as we explained. One of the original objectives of our study was to compare the exposure of children and adolescents to HFSS food advertising to that of adults. Nevertheless, data on adults were not sufficiently exploited in the manuscrit.
We deleted all results about adults as you suggested, in order to clarify the purpose.
Lines 95-96. In my opinion the sentence ”An overview of food advertising investments and media usage times provides the background for the justification and progress of the study” needs to be rewritten.
>>The sentence was indeed too allusive. We proposed the following reformulation:
An overview of food advertising investments and media usage times allowed us to identify in advance the media on which the most advertisements were broadcast and the media most used by children and adolescents.
The section 2.1. ‘Source of data’ should be rewritten due to an excessive amount of information. The main objective of the study was to quantify the exposure of French children and adolescents to television advertising and TV sponsorship for food products. Thus, information about Kantar Media research is not needed. For example: ‘This company identified the advertisements broadcast on television, in the cinema, on the radio, in the press, or by outdoor advertising and, in part, advertisements on the Internet’. I understood that the television advertisement is in the scope of the presented results of analysis. Moreover, information on ‘data relating to radio come from the Médiamétrie 126,000 survey, a telephone survey with quota sampling conducted each year among 126,000 people.. “ leads to misunderstanding. Short information on the data from these surveys used in the analysis will be satisfying.
>> We agree that the source of data section was very comprehensive and detailed. Numerous sources of complementary information were used for this study and we selected a limited number of relevant results for the purpose of this manuscript. Following this comment, we simplified the section so that it does not lead to misunderstanding.
Lines 257 – 262 The Figure 1 is not necessary in the manuscript. It presents only two numbers which are described earlier.
>> Following this comment, we deleted the figure 1.
Line 266 In my opinion the word ‘media’ is not necessary in this sentence.
>> The sentence was rewritten as follows :
In 2018, children and adolescents spent the most time per day on TV and on the Internet
Figure 3, 4, and 5 – the same colors should be used to show the audience per time slot.
>> The colors were modified and figure about adults was deleted.
Lines 318-322 The section 3.4.1 is too short to be identified.
>> Titles of sections 3.4.1 and 3.4.2 were deleted. An introductive sentence was added in the manuscript.
The section 3.4.2. ‘Percentage of television advertisements for DE Nutri-Score products viewed according to the audience of children and adolescents’ needs improvement.
>> We completed the section by adding the general direction of the results presented:
In view of our results regarding the TV audience according to time slot, three audience thresholds were considered: 10%, 15%, and 20%, showing that the more viewing time slots considered, the higher the percentage of ads for Nutri-Score products seen by children and adolescents.
Figure 6 an 7 duplicate the information in the text and vice versa. The titles of figures and explanation sound a little strange. I suggest to delete the figures from the text.
>> As the results presented combine several notions that are not self-evident to identify, we suggest not to delete these figures that allow to show the correspondence between audience thresholds and timeslots.
In Discussion section there are a lot of results. Such repetition is not appropriate.
>> We deleted some results and reduced this section.
I suggest to rewrite the conclusion – more references to the results of the own study.
>> We agree that the conclusion did not refer enough to the results of the study. We completed it with the main results of the study.
There are many editorial errors in the text, e.g. unnecessary dots, information in parentheses etc.
We deleted the majority of the information in parentheses
Some examples:
Lines 5-10. Capital letters in the names of institutions
>> Capital letters in the names of institutions can’t be used in some cases. After confirmation with the editor of our direction, for Santé publique France, as for any institute in general, there is no typographical reason to capitalize an adjective that follows a noun. On the other hand, France is a country name which justifies the use of a capital letter.
Line 19 instead of ‘ should be .
>> It was a typing error. Thank you for your vigilance.
Line 61 1 in my opinion it would be better to present short definition of communication in the text.
>> This definition of commercial communications is a juridical one, that can not easily be simplified. We simplified it and added a reference for the simplified definition. As it is in the introduction, we suggest to keep it in a footnote.
Reviewer 2 Report
Introduction
Seems sparse and the hanging paragraphs needs to be revised. Please only include relevant topic and make it more concise.
Section 2.3
There are numerous hypotheses that has been proposed, can the authors postulate a cause and effect model here?
This Section needs to be clarified what are the intention of the study, does it answer the questions proposed Introduction? Right now it feels that the authors are not focused here.
Results
This is very descriptive and feels like just an output from the Kantar platform, did the authors looks and attempt further analysis for a cause and effect model here?
Discussion
What are the key results here? The authors needs to run more analysis to develop a cause-effect model otherwise this study would be more descriptive and not too useful by its own.
Comment
English needs some revision, some sentences doesn't flow well.
Author Response
Introduction
Seems sparse and the hanging paragraphs needs to be revised. Please only include relevant topic and make it more concise.
We thank the reviewer for their feedback on our manuscript. Following this comment, the introduction section was modified, as follows:
We deleted some too detailed informations. We have also rewritten some passages to show the links between the topics discussed: obesity is a multifactorial disease, excessive calories ingested and excessive consumption of high in fat, salt and sugar foods (HFSS products) are contributing factors of obesity in particular that of children and adolescents, food marketing has an impact on children and adolescents unhealthy food intake. This calls for public health action. Conducting studies to monitor the marketing of unhealthy products to children and adolescents sustains the implementation of regulations and restrictions by States. The study presented in the manuscrit contributes to French monitoring of the exposure of French children and adolescents to advertising for foods high in fat, sugar or salt
Section 2.3
There are numerous hypotheses that has been proposed
This Section needs to be clarified what are the intention of the study, does it answer the questions proposed Introduction? Right now it feels that the authors are not focused here.
The two objectives of the study were the quantification of the exposure of children and adolescents to (1) advertising for HFSS foods and to (2) youth TV programs. Following this comment. We removed non-essential intermediate variables to clarify the purpose.
Can the authors postulate a cause and effect model here?
>> Our approach was purely descriptive, to provide important context elements regarding advertising to children in France.
Our study contributes to the monitoring of the marketing of unhealthy products to children and adolescents, which is recommended by WHO regional office for Europe.
The study is also ultimately aimed to provide policy makers efficient indicators to regulate TV advertising.
It uses specific types of data - audience data and advertisements broadcast data - which are not of the same nature as individual behavioural data. In our opinion, it would not have been possible for this reason, to develop a cause effect model. Furthermore, individual data on obesity and individual factors contributing to obesity are not available in this study.
The method used for this descriptive study was based on the one of a European study on the exposure of children to linear, non-linear and online marketing of foods high in fat, salt or sugar, commissioned by the European Commission.
This European study was commissioned as part of the European Joint Action on implementation of validated best practices on nutrition (JA Best-ReMaP, 2020), one of the objectives of which is to develop a harmonized European approach to reduce the exposure of children and adolescents to food marketing and to use common tools to monitor this exposure. France is participating in the joint action.
We added this precision in the manuscript.
Results
This is very descriptive and feels like just an output from the Kantar platform
>> While the study is only descriptive, it does built on a number of sources of data that were compiled to obtain the results, beyond Kantar data.
First, in order to get results comparable to those of the European reference study on the subject, the audience of specific Médiamétrie age groups were analysed
Secondly, to quantify the exposure of children and adolescents to advertisements for HFSS foods, specific analyses, that were not only an output from the Kantar platform, were performed. They were carried out using the Mxplorer software, specific media performance analysis software.
We added in the method the following precision about the analyses performed:
TV food advertisements coded according to the Nutri-Score and identified according to the days and times of broadcast, were cross-referenced with audience data for 4- to 12-year-olds and 13- to 17-year-olds. An analysis of this cross-referencement by time slot of children and adolescents audience has then been performed.
Did the authors looks and attempt further analysis for a cause and effect model here?
>> See above our answer to the last question in Section 2.3 section
Discussion
What are the key results here? The authors needs to run more analysis to develop a cause-effect model otherwise this study would be more descriptive and not too useful by its own.
>> Following this comment, the discussion was reorganized to allow a more fluid presentation of the main results and their contribution to the current body of literature on the subject of TV advertising to children.
An introductory paragraph was added to summarize the public health issues, in this case the regulation of advertising, to which the results of this descriptive study provide some guidance. The usefulness of this study to provide indicators for the implementation of such a measure to health policy makers is explained. The key results obtained in the different stages of the study were better highlighted in the overall discussion.
>> Point about cause-effect model: see above our answer to the last question in Section 2.3 section
>> Descriptive studies as the present one are particularly useful in the specific context of food marketing. As mentioned by the WHO regional office for Europe monitoring the marketing of unhealthy products to children and adolescents is fundamental to gather evidence to support policy deliberations to implement regulations and restrictions by states. With respect to TV food marketing, the monitoring aims to answer the following questions, supported by descriptive studies:
- How much food and beverage advertising are children (under 18 years of age) likely to be exposed to on TV?
- What foods and beverages are advertised?
Comment
English needs some revision, some sentences doesn't flow well.
>> A translation was done by a professional. We simplified some sentences.
Round 2
Reviewer 1 Report
Dear Authors, I appreciate all improvements made. Please consider the possibility of combining figures 4 and 5.Kind regards
Reviewer
Reviewer 2 Report
The authors have addressed the comments.